# Lithocholic Acid Amides as Potent Vitamin D Receptor Agonists

**DOI:** 10.3390/biom12010130

**Published:** 2022-01-14

**Authors:** Ayana Yoshihara, Haru Kawasaki, Hiroyuki Masuno, Koki Takada, Nobutaka Numoto, Nobutoshi Ito, Naoya Hirata, Yasunari Kanda, Michiyasu Ishizawa, Makoto Makishima, Hiroyuki Kagechika, Aya Tanatani

**Affiliations:** 1Department of Chemistry, Faculty of Science, Ochanomizu University, 2-1-1 Otsuka, Bunkyo-ku, Tokyo 112-8610, Japan; yoshihara.ayana.3348@gmail.com (A.Y.); cindy228.hr73@i.softbank.jp (H.K.); 2Institute of Biomaterials and Bioengineering, Tokyo Medical and Dental University (TMDU), 2-3-10 Kanda-Surugadai, Chiyoda-ku, Tokyo 101-0062, Japan; masuno.chem@tmd.ac.jp; 3Medical Research Institute, Tokyo Medical and Dental University (TMDU), 1-5-45 Yushima, Bunkyo-ku, Tokyo 113-8510, Japan; 170221ds@tmd.ac.jp (K.T.); numoto.str@mri.tmd.ac.jp (N.N.); ito.str@tmd.ac.jp (N.I.); 4National Institute of Health Sciences, 3-25-26 Tonomachi, Kawasaki-ku, Kawasaki 210-9501, Japan; n-hirata@nihs.go.jp (N.H.); kanda@nihs.go.jp (Y.K.); 5Department of Biomedical Sciences, Nihon University School of Medicine, 30-1 Ohyaguchikamimachi, Itabashi-ku, Tokyo 173-8610, Japan; ishizawa.michiyasu@nihon-u.ac.jp (M.I.); makishima.makoto@nihon-u.ac.jp (M.M.)

**Keywords:** vitamin D, nuclear receptor, lithocholic acid, amide, cell differentiation

## Abstract

1α,25-Dihydroxyvitamin D_3_ [1α,25(OH)_2_D_3_, **1**] is an active form of vitamin D_3_ and regulates various biological phenomena, including calcium and phosphate homeostasis, bone metabolism, and immune response via binding to and activation of vitamin D receptor (VDR). Lithocholic acid (LCA, **2**) was identified as a second endogenous agonist of VDR, though its potency is very low. However, the lithocholic acid derivative **3** (**Dcha-20**) is a more potent agonist than 1α,25(OH)_2_D_3_, (**1**), and its carboxyl group has similar interactions to the 1,3-dihydroxyl groups of **1** with amino acid residues in the VDR ligand-binding pocket. Here, we designed and synthesized amide derivatives of **3** in order to clarify the role of the carboxyl group. The synthesized amide derivatives showed HL-60 cell differentiation-inducing activity with potency that depended upon the substituent on the amide nitrogen atom. Among them, the *N*-cyanoamide **6** is more active than either **1** or **3**.

## 1. Introduction

Vitamin D receptor (VDR) is a ligand-dependent transcriptional factor belonging to the nuclear receptor superfamily [1,2], and mediates most of the biological functions of vitamin D_3_, including calcium and phosphate homeostasis, bone metabolism, and immune regulation. The endogenous agonist of VDR is 1α,25-dihydroxyvitamin D_3_ [1α,25(OH)_2_D_3_, **1**], an active metabolite of vitamin D_3_ (Figure 1), which induces increased expression of target genes. Various vitamin D derivatives have been synthesized as candidate drugs for skin and bone diseases [3,4,5], though most of them have the same secosteroid structure as 1α,25(OH)_2_D_3_ (**1**).

Lithocholic acid (LCA, **2**, Figure 1) is a secondary bile acid formed from chenodeoxycholate, and was identified as a second endogenous agonist of VDR [6,7,8,9]. However, its VDR binding affinity and potency are very low, compared with those of **1**. We recently developed a potent lichocholic acid derivative **3** (**Dcha-20**) that has a 2-hydroxy-2-methylprop-1-yl moiety instead of the 3-hydroxyl group at the 3 α position of **2** (Figure 1) [10]. In HL-60 cell differentiation-inducing assay, **3** was more potent than **1**. Gaikwad, S. et al. recently reported that **3** shows potent vitamin D activity with a lower calcemic activity than **1** [11].

Analysis of the crystal structure of VDR ligand-binding domain (LBD) bound to **3** (**Dcha-20**) showed that the 3-substituent forms direct hydrogen bonds with the two histidine residues His301 and His393, like the 25-hydroxyl group of 1α,25(OH)_2_D_3_ (**1**), but different from the indirect interaction, via a water molecule, of the 3-hydroxyl group of LCA (**2**) with the same amino acid residues of the VDR [8,11]. The carboxyl group of **3** forms hydrogen bonds with Tyr143 and Ser274, as in the case of 1α,25(OH)_2_D_3_ (**1**) or LCA (**2**). The carboxyl group of LCA (**2**) also formed indirect hydrogen bonds with Arg270 and Ser233 via a water molecule, while the corresponding interaction was not observed in the case of **3**. The 3-hydroxyl group of 1α,25(OH)_2_D_3_ (**1**) forms direct hydrogen bonds with these amino acid residues in the crystal. Further, our preliminary results on the pharmacokinetics of **3** indicated that this compound is eliminated very quickly from the serum in mice (data not shown), and the carboxyl group is one of the target functional groups for improvement of this undesirable feature. Therefore, in this study, we designed and synthesized several lithocholic acid amide derivatives **4**–**8** with various *N*-substituents in order to clarify the role of the carboxyl group in the VDR binding and vitamin D activity of **3** (Figure 2).

## 2. Results and Discussion

### 2.1. Synthesis

The lithocholic acid amide derivatives **4**–**8** were synthesized from LCA (**2**) via compound **3** (**Dcha-20**), which was prepared by modification of our previous method [10] (Figure 1). Acetylation of the 3-hydroxyl group of LCA (**2**), followed by reduction of the carboxyl group and benzylation, afforded compound **11**. After hydrolysis of the acetate and oxidation, the 3-keto compound **13** was converted to a mixture of 3-formyl derivatives (**15a**:**15b** = 8:2) via Wittig reaction. Treatment of this mixture with potassium carbonate increased the ratio of the desired 3 α-isomer **15a** (**15a**:**15b** = 10:1).

Compound **3** was synthesized from aldehyde **15a** in 7 steps according to our reported method (Figure 2). Briefly, **15a** was reduced to the 3-hydroxymethyl compound **16**, followed by tosylation and reaction with sodium cyanide, to afford the nitrile **18**. Two-step methylation of **18** gave compound **20** with a 2-hydroxy-2-methylprop-1-yl moiety at the 3 position. The terminal polar group in the side chain of **20** was converted to a carboxyl group in 2 steps to afford **3** (**Dcha-20**). The amide derivatives **4**–**8** were synthesized from **3** by the method shown in Figure 3.

### 2.2. Biological Evaluation

The vitamin D activity of the synthesized lithocholic acid amide derivatives was evaluated in terms of cell differentiation-inducing activity toward human acute promyelocytic leukemia cell line HL-60 [12]. HL-60 cell differentiation was evaluated in terms of the ratio of nitroblue tetrazolium (NBT)-positive cells (Figure 3 and Table 1). All the amide derivatives examined induced dose-dependent differentiation of HL-60 cells. In this assay, the unsubstituted amide **4a** exhibited more potent activity (EC_50_: 0.44 nM), compared with that of the carboxylic acid derivative **3** (**Dcha-20**, EC_50_: 1.01 nM) or 1α,25(OH)_2_D_3_ (**1**, EC_50_: 0.74 nM). Interestingly, *N*-monomethylation of compound **4a**, yielding compound **4b**, diminished the activity. *N*-Methyl group would disturb the hydrogen bond formation of amide group with the amino acid residues of VDR. The introduction of an *N*-hydroxyl (compound **5a**, EC_50_: 1.18 nM) or *N*-methoxyl group (compound **5b**, EC_50_: 1.45 nM) slightly decreased the activity, though these compounds still showed activity comparable to that of **3**. Interestingly, compound **6** bearing an *N*-cyano group (EC_50_: 0.32 nM) was the most active among the synthesized amide derivatives, being more potent than **3** or **1**.

Among the three derivatives bearing an *N*-carboxyalkyl group, compound **7a** with one methylene group between the amide and carboxyl groups (EC_50_: 2.03 nM) showed lower activity than the parent amide compound **4a** (EC_50_: 0.44 nM), while compounds **7b** (EC_50_: 0.45 nM) and **7c** (EC_50_: 0.64 nM) with longer alkyl chains were more active than **7a**. A similar tendency was observed for the compounds bearing an *N*-sulfoalkyl group. Thus, compound **8b** (EC_50_: 6.56 nM) was more active than compound **8a** (EC_50_: 18.5 nM). Terminal polar groups (carboxyl for **7** and sulfo for **8**) adjacent to the amide group appear to have a negative effect, possibly blocking hydrogen bond formation of the amide with amino acid residues of VDR, whereas groups more distant from the amide group might have positive effects such as formation of additional hydrogen bond(s).

Next, we examined the VDR trasactivation ability for selected compounds (Figure 4 and Table 1), according to the method reported in our previous study [13]. 1α,25(OH)_2_D_3_ (**1**, EC_50_: 0.058 nM) and compound **3** (**Dcha-20**, EC_50_: 0.083 nM) showed potent transactivation activity at the concentrations above 0.1 nM, while LCA (**2**) did not show the activity at the concentration below 1 μ M. All the amide derivatives examined showed dose-dependent transactivation activity, which was well correlated with the activity in HL-60 cell assay. Among them compounds **6** (EC_50_: 0.10 nM) and **7c** (EC_50_: 0.081 nM) were as potent as 1α,25(OH)_2_D_3_ (**1**) and compound **3** (**Dcha-20**). The results indicated that the differentiation-inducing activity of the lithocholic acid amide derivatives would be mediated by VDR.

### 2.3. X-ray Crystallographic Analysis

We next attempted X-ray crystallographic analysis of the complex of rat VDR LBD (residues 116–423, Δ165–211) with several of the lithocholic acid amide derivatives. According to the method reported in our previous study [10,14], a synthetic peptide containing the target sequence of the coactivator MED1 (mediator of RNA polymerase II transcription subunit 1, also known as ARC205 or DRIP205) was included in the crystallization solution of VDR LBD and the test compound. However, the analysis was successful only for the complex of compound **7b** (Table 2). The electron density map clearly shows the VDR LBD, the coactivator peptide, the ligand and a relatively low number of water molecules. Figure 5a shows the overall structure of the VDR LBD complex with **7b**; it is similar to those previously reported for VDR LBD complexes with other lithocholic acid derivatives [10,14,15]. The interactions of compound **7b** with amino acid residues of the VDR LBD (Figure 5b) are compared with those of **3** (**Dcha-20**) in Figure 5c. The hydroxyl group in the 3-substituent of **7b** forms direct hydrogen bonds with two histidine residues, His301 (O···N distance: 2.79 Å) and His393 (O···N distance: 2.68 Å). This is the same as in the case of **3** (**Dcha-20**), in which the O···N distances were 2.80 Å for His301 and 2.66 Å for His393, whereas LCA (**2**) forms indirect hydrogen bonds with these amino acid residues via a water molecule. The direct interactions of the hydroxy group in the 3-substituent with two histidines may contribute to the potent activity of **7b** and **3**. The carboxyl group of compound **3** (**Dcha-20**) formed hydrogen bonds with the phenolic hydroxyl group of Tyr143 (O··O distance: 2.81 Å) in helix 1 and the hydroxymethyl group of Ser274 (O··O distance: 3.13 Å) in helices 4/5, whereas the amide group of **7b** did not form a hydrogen bond with any amino acid residue. Instead, the terminal carboxyl group of **7b** formed hydrogen bonds with Arg270 (O···N distance: 2.82 Å) and the backbone amide bond of Tyr143 (O···N distance: 2.91 Å). Similar hydrogen bond formation with these amino acid residues of the VDR LBD was observed in secosteroid derivatives bearing a hydroxylated substituent at the 2-position of the cyclohexane ring.

## 3. Conclusions

We designed and synthesized several lithocholic acid amide derivatives of **3** (**Dcha-20**) as a lead compound. The carboxyl group of compound **3** (**Dcha-20**) can be replaced with various amide bonds without decreasing the activity. Among the synthesized amide derivatives, compounds **4a**, **6** with an *N*-cyano group and **7b** with an *N*-2-carboxyethyl group showed the most potent activity. Crystallographic analysis of the complex of the VDR LBD with **7b** showed that the terminal carboxyl group, but not the amide group, forms hydrogen bonds with amino acid residues of the VDR LBD. The hydroxyl group in the 3-substituent also forms direct hydrogen bonds with two histidine residues, His301 and His393. Recently, compound **3** (**Dcha-20**) was reported to have lower calcemic activity than 1α,25(OH)_2_D_3_ (**1**) [11], which would be favorable for clinical application, but preliminary studies on the pharmacokinetics of **3** (**Dcha-20**) indicated that it is eliminated very quickly in mice. The carboxyl group of **3** (**Dcha-20**) appears to be important for both the potent vitamin D activity and the pharmacokinetic properties. The novel amide derivatives of compound **3** also showed potent vitamin D activities, and studies on their pharamacokinetic properties are now on going. Our results suggest that it may be possible to develop lithocholic acid derivatives having potent activity and drug-like pharmacokinetic properties by chemical modification at the terminal polar group in the side chain.

## 4. Experimental

### 4.1. General

^1^H and ^13^C NMR spectra were recorded on JNM-ECS 400, JNM-ECS 500, and Bruker Avance 600 spectrometers. The ^1^H NMR chemical shifts are reported in parts per million (ppm) relative to the centerline of the singlet signal of the solvent molecule (7.26 ppm for chloroform); coupling constants are given in hertz (Hz). The ^1^^3^C NMR chemical shifts are reported in ppm relative to the centerline of the triplet at 77.16 ppm for CDCl_3_. Mass spectral data were obtained on a Bruker Daltonics micro TOF-2focus in the positive and negative ion detection modes.

### 4.2. Synthesis

**Synthesis of compound 9**: Acetic anhydride (6.174 g, 60.48 mmol) and 4-dimethylaminopyridine (97 mg, 0.80 mmol) were added to a solution of lithocholic acid (1.522 g, 4.04 mmol) in dry pyridine (40 mL). The mixture was stirred for 20 h at room temperature, then quenched with water and extracted with a mixture of ethyl acetate and *n*-hexane (1:1). The organic layer was washed with 2 M hydrochloric acid and brine, dried over sodium sulfate, and filtered. The filtrate was concentrated to give **9** (1.757 g, quant.) as a yellow solid. ^1^H NMR (600 MHz, CDCl_3_) δ 4.74-4.70 (m, 1 H), 2.40 (ddd, *J* = 15.6, 10.2, 5.4 Hz, 1 H), 2.26 (ddd, *J* = 15.6, 9.6, 6.6 Hz, 1 H), 2.03 (s, 3 H), 1.98-1.95 (m, 1 H), 1.85-1.80 (m, 5 H), 1.68-1.00 (m, 20 H), 0.92 (d, *J* = 6.6 Hz, 3 H), 0.92 (s, 3 H), 0.64 (s, 3 H); ^13^C NMR (150 MHz, CDCl_3_) δ 178.33, 170.73, 74.39, 56.43, 55.90, 42.70, 41.82, 40.33, 40.08, 35.72, 35.28, 34.98, 34.54, 32.18, 30.71, 30.60, 28.16, 26.96, 26.57, 26.27, 24.14, 23.30, 21.50, 20.78, 18.20, 12.01; HRMS calcd for C_26_H_42_NaO_4_ (M + Na)^+^ 441.2975, found 441,2970.

**Synthesis of compound 10**: Triethylamine (533 mg, 5.27 mmol) and ethyl chloroformate (616 mg, 5.680 mmol) were added to a solution of **9** (1.757 g, 4.04 mmol) in distilled THF (40 mL). The mixture was stirred for 2 h at room temperature, then cooled to 0 °C, and sodium borohydride (737 mg, 19.5 mmol) and dry methanol (20 mL) were added to it. The reaction mixture was stirred for 2 h 15 min at 0 °C and then quenched with water. After removal of the solvent in vacuo, the residue was extracted with ethyl acetate. The organic layer was washed with brine, dried over sodium sulfate, filtered, and concentrated. The residue was purified by silica gel column chromatography (ethyl acetate/*n*-hexane = 1:3) to give **10** (1.672 g, quant.) as a colorless oil. ^1^H NMR (600 MHz, CDCl_3_) δ 4.74-4.69 (m, 1 H), 3.65-3.58 (m, 2 H), 2.03 (s, 3 H), 1.99-1.96 (m, 1 H), 1.97-1.80 (m, 4 H), 1.70-1.02 (m, 23H), 0.93 (s, 3 H), 0.92 (d, *J* = 6.6 Hz, 3 H), 0.65 (s, 3 H); ^13^C NMR (150 MHz, CDCl_3_) δ 170.68, 74.37, 63.58, 56.43, 56.10, 42.62, 41.81, 40.32, 40.08, 35.70, 35.51, 34.95, 34.51, 32.16, 31.74, 29.31, 28.25, 26.95, 26.55, 26.26, 24.13, 23.28, 21.47, 20.76, 18.57, 11.98; HRMS calcd for C_26_H_44_NaO_3_ (M + Na)^+^ 427.3183, found 427.3174.

**Synthesis of compound 11**: Trifluoromethanesulfonic acid (0.25 mL, 2.85 mmol) was added to a solution of **10** (1.309 g, 3.24 mmol) in dry 1,4-dioxane (40.0 ml) and benzyl 2,2,2-trichloroacetimidate (2.206 g, 8.74 mmol) at 0 °C under an argon atmosphere. The reaction mixture was stirred for 3 h at room temperature, then cooled to 0 °C, quenched with saturated sodium hydrogen carbonate, and extracted with ethyl acetate. The organic layer was washed with brine, dried over sodium sulfate, filtered, and concentrated. The residue was purified by silica gel column chromatography (ethyl acetate/*n*-hexane = 1:14) to give compound **11** (1.297 g, 81%) as a colorless oil. ^1^H NMR (600 MHz, CDCl_3_) δ 7.35-7.32 (m, 4 H), 7.30-7.27 (m, 1 H), 4.74-4.70 (m, 1 H), 4.51 (d, *J* = 12.0 Hz, 1 H), 4.49 (d, *J* = 12.0, 1 H), 3.46-3.40 (m, 2 H), 2.03 (s, 3 H), 1.98-1.96 (m, 1 H), 1.84-1.79 (m, 4 H), 1.68-1.66 (m, 2 H), 1.56-1.02 (m, 21 H), 0.92 (s, 3 H), 0.91 (d, *J* = 6.6 Hz, 3 H), 0.63 (s, 3 H); ^13^C NMR (150 MHz, CDCl_3_) δ 170.71, 138.61, 128.30, 127.60, 127.43, 74.41, 72.29, 71.01, 56.44, 56.14, 42.63, 41.83, 40.33, 40.10, 35.73, 35.53, 34.97, 34.53, 32.18, 32.12, 28.24, 26.98, 26.56, 26.28, 26.25, 24.16, 23.30, 21.48, 20.77, 18.55, 11.99; HRMS calcd for C_33_H_50_NaO_3_ (M + Na)^+^ 517.3652, found 517.3638.

**Synthesis of compound 12**: Potassium carbonate (1.464 g, 10.6 mmol) was added to a solution of **11** (1.344 g, 2.72 mmol) in dry methanol (30 mL) and distilled THF (5 mL). The mixture was stirred for 6 h 20 min at room temperature under an argon atmosphere and then quenched with acetic acid. After removal of the solvent in vacuo, the residue was extracted with ethyl acetate. The organic layer was washed with water and brine, dried over sodium sulfate, filtered, and concentrated. The residue was purified by silica gel column chromatography (ethyl acetate/*n*-hexane = 1:3) to give **12** (1.190 g, 97%) as a colorless oil. ^1^H NMR (600 MHz, CDCl_3_) δ 7.36-7.33 (m, 4 H), 7.30-7.27 (m, 1 H), 4.51 (d, *J* = 13.2 Hz, 1 H), 4.49 (d, *J* = 12.0 Hz, 1 H), 3.64-3.60 (m, 1 H), 3.46-3.40 (m, 2 H), 1.96 (dt, *J* = 12.0, 3.0 Hz, 1 H), 1.86-1.66 (m, 7 H), 1.59-0.94 (m, 20 H), 0.91 (s, 3 H), 0.91 (d, *J* = 6.6 Hz, 3 H), 0.63 (s, 3 H); ^13^C NMR (150 MHz, CDCl_3_) δ 138.65, 128.32, 127.61, 172.43, 72.80, 71.88, 71.02, 56.46, 56.13, 42.64, 42.04, 40.37, 40.13, 36.39, 35.79, 35.57, 35.29, 34.53, 32.13, 30.50, 28.27, 27.16, 26.39, 26.28, 24.20, 23.35, 20.78, 18.56, 12.00; HRMS calcd for C_31_H_48_NaO_2_ (M + Na)^+^ 475.3547, found 475.3541.

**Synthesis of compound 13**: Sulfuric acid (0.23 mL) was added to a cooled solution of chromium (VI) oxide (267 mg) in water (0.77 mL) just prior to use. An aliquot (0.7 mL) of this Jones reagent was added to a solution of **12** (1.173 g, 2.59 mmol) in dry acetone (30 mL). The mixture was stirred for 30 min at room temperature, then quenched with 2-propanol, and the solvent was removed in vacuo. The extract was extracted with diethyl ether. The organic layer was washed with brine, dried over sodium sulfate, filtered, and concentrated. The residue was purified by silica gel column chromatography (ethyl acetate/*n*-hexane = 1:4) to give **13** (1.148 g, 98%) as a colorless oil. ^1^H NMR (600 MHz, CDCl_3_) δ 7.36-7.33 (m, 4 H), 7.29-7.27 (m, 1 H), 4.51 (d, *J* = 12.6 Hz, 1 H), 4.49 (d, *J* = 12.6 Hz, 1 H), 3.47-3.40 (m, 2 H), 2.70 (t, *J* = 14.4 Hz, 1 H), 2.33 (td, *J* = 14.4, 5.4 Hz, 1 H), 2.17-2.14 (m, 1 H), 2.05-2.00 (m, 3 H), 1.88-1.79 (m, 3 H), 1.70-1.08 (m, 19 H), 1.01 (s, 3 H), 0.92 (d, *J* = 6.6 Hz, 3 H), 0.67 (s, 3 H); ^13^C NMR (150 MHz, CDCl_3_) δ 213.64, 138.62, 128.31, 127.60, 127.44, 72.81, 70.99, 56.40, 56.12, 44.32, 42.68, 42.35, 40.65, 40.01, 37.21, 36.99, 35.54, 35.48, 34.85, 32.12, 28.22, 26.58, 26.17, 25.74, 24.14, 22.63, 21.15, 18.57, 12.03; HRMS calcd for C_31_H_46_NaO_2_ (M + Na)^+^ 473.3390, found 473.3379.

**Synthesis of compound 14**: A mixture of (methoxylmethyl)triphenylphosphonium chloride (18.111 g, 52.8 mmol) and potassium *tert*-butoxide (6.188 g, 46.7 mmol) in distilled THF (112 mL) was stirred for 45 min at 0 °C under an argon atmosphere. A solution of **13** (6.812 g, 15.1 mmol) in distilled THF (18 mL) was added to it. The resulting mixture was allowed to warm to room temperature, stirred for 2 h, quenched with water, and extracted with ethyl acetate. The organic layer was washed with water and brine, dried over sodium sulfate, filtered, and concentrated. The residue was purified by silica gel column chromatography (chloroform/*n*-hexane = 3:2) to give **14** (7.280 g, quant.) as a mixture of geometrical isomers. Each isomer was isolated in a small amount to determine the structure. (*E*)-**14**: ^1^H NMR (600 MHz, CDCl_3_) δ 7.35-7.33 (m, 4 H), 7.29-7.27 (m, 1 H), 5.76 (s, 1 H), 4.51 (d, *J* = 12.0 Hz, 1 H), 4.49 (d, *J* = 12.0 Hz, 1 H), 3.52 (s, 3 H), 3.46-3.40 (m, 2 H), 2.31 (dd, *J* = 13.8, 3.6 Hz, 1 H), 2.12 (t, *J* = 13.8 Hz, 1 H), 1.98-1.95 (m, 1 H), 1.86-1.79 (m, 3 H), 1.71-1.66 (m, 2 H), 1.57-0.92 (m, 20 H), 0.91 (s, 3 H), 0.91 (d, *J* = 5.4 Hz, 3 H), 0.64 (s, 3 H); ^13^C NMR (150 MHz, CDCl_3_) δ 138.75, 138.63, 128.32, 127.61, 127.43, 118.80, 72.82, 71.07, 59.28, 56.60, 56.23, 43.67, 42.72, 40.24, 40.15, 38.68, 35.76, 35.59, 32.22, 28.29, 26.99, 26.34, 26.28, 25.71, 25.12, 24.24, 23.73, 20.97, 18.61, 12.06; HRMS calcd for C_33_H_50_NaO_2_ (M + Na)^+^ 501.3707, found 501.3698. (*Z*)-**14**: ^1^H NMR (600 MHz, CDCl_3_) δ 7.35-7.33 (m, 4 H), 7.29-7.27 (m, 1 H), 5.72 (s, 1 H), 4.51 (d, *J* = 12.0 Hz, 1 H), 4.49 (d, *J* = 12.0 Hz, 1 H), 3.52 (s, 3 H), 3.46-3.40 (m, 2 H), 2.46 (d, *J* = 13.8 Hz, 1 H), 2.39 (t, *J* = 13.6 Hz, 1 H), 1.98-1.95 (m, 1 H), 1.86-1.79 (m, 3 H), 1.71-1.66 (m, 2 H), 1.57-0.92 (m, 20 H), 0.91 (s, 3 H), 0.91 (d, *J* = 5.4 Hz, 3 H), 0.64 (s, 3 H). 13C; HRMS calcd for C_33_H_50_NaO_2_ (M + Na)^+^ 501.3707, found 501.3698.

**Synthesis of compound 15**: 6 M hydrochloric acid (30 mL) was added to a solution of **14** (6.975 g, 14.6 mmol) in distilled THF (80 mL), and the mixture was stirred for 5 h at room temperature. The reaction mixture was quenched with water, and extracted with ethyl acetate. The organic layer was washed with water and brine, dried over sodium sulfate, filtered, and concentrated. The residue was purified by silica gel column flash chromatography (dichloromethane/*n*-hexane = 1:1) to give a mixture of **15a** and **15b** (6.664 g, 98%, **15a**:**15b** = 8:2) as a colorless oil. **15a**: ^1^H NMR (600 MHz, CDCl_3_) δ 9.64 (d, *J* =1.8 Hz, 1 H), 7.36-7.33 (m, 4 H), 7.29-7.27 (m, 1 H), 4.51 (d, *J* = 13.2 Hz, 1 H), 4.49 (d, *J* = 12.0 Hz, 1 H), 3.46-3.40 (m, 2 H), 2.27 (dtt, *J* = 12.6, 3.6, 1.8 Hz, 1 H), 1.97-0.97 (m, 28 H), 0.96 (s, 3 H), 0.91 (d, *J* = 6.6 Hz, 3 H), 0.63 (s, 3 H); ^13^C NMR (150 MHz, CDCl_3_) δ 205.13, 138.64, 128.31, 127.61, 127.43, 72.80, 71.01, 56.36, 56.11, 51.32, 42.62, 42.56, 40.32, 40.03, 36.06, 35.74, 35.56, 35.10, 32.12, 28.24, 27.16, 26.28, 24.16, 23.81, 20.78, 18.55, 12.00; HRMS calcd for C_32_H_48_NaO_2_ (M + Na)^+^ 487.3547, found 487.3535. **15b**: ^1^H NMR (600 MHz, CDCl_3_) δ 9.71 (s, 1 H), 7.36-7.33 (m, 4 H), 7.23-7.27 (m, 1 H), 4.51 (d, *J* = 12.6 Hz, 1 H), 4.49 (d, *J* = 12.6 Hz, 1 H), 3.46-3.39 (m, 2 H), 2.45 (br, 1 H), 2.04-0.85 (m, 28 H), 0.91 (d, *J* = 6.6 Hz, 3 H), 0.87 (s, 3 H), 0.63 (s, 3 H); ^13^C NMR (150 MHz, CDCl_3_) δ 206.55, 138.64, 128.33, 127.63, 127.46, 72.82, 71.03, 56.56, 56.16, 47.08, 42.68, 40.15, 39.98, 39.60, 35.58, 35.54, 34.90, 33.48, 32.14, 28.26, 26.91, 26.26, 26.07, 25.00, 24.18, 23.85, 20.85, 19.60, 18.58, 12.02; HRMS calcd for C_32_H_48_NaO_2_ (M + Na)^+^ 487.3547, found 487.3531.

The ratio of **15a** in the mixture of the epimers was increased by treatment of the mixture with K_2_CO_3_, MeOH, THF (66%), and the isolated **15a** was converted to compound **3** (**Dcha-20**), according to our reported method.

**Synthesis of compound 20**: Acetyl chloride (0.01 mL, 0.140 mmol) was added to a cooled solution of **3** (64 mg, 0.15 mmol) in methanol (7 mL) at 0 °C under an argon atmosphere. The mixture was stirred for 4 h at room temperature, then quenched with water at 0 °C, and the precipitate was collected to obtain **20** (72 mg, quant.) as a colorless solid. ^1^H NMR (400 MHz, CDCl_3_) δ 3.66 (s, 3H), 2.39-2.31 (m, 1H), 2.25-2.17 (m, 1H), 1.94 (d, *J* = 11.4 Hz, 1H), 1.89-1.73 (m, 4H), 1.59-0.90 (m, 25H), 1.22 (s, 6H), 0.91 (d, *J* = 6.4 Hz, 3H), 0.91 (s, 3H), 0.64 (s, 3H); ^13^C NMR (150 MHz, CDCl_3_) δ 174.84, 71.65, 60.39, 56.50, 55.89, 51.50, 51.23, 43.55, 42.70, 40.46, 40.15, 37.43, 35.89, 35.80, 35.34, 34.97, 34.68, 31.01, 30.97, 30.02, 29.97, 28.18, 27.40, 26.45, 24.17, 23.97, 21.07, 20.78, 18.24, 14.17, 12.01; HRMS calcd for C_29_H_50_NaO_3_ (M + Na)^+^ 469.3652, found 484.3.655.

**Synthesis of compound 4**: A methanolic solution of conc. ammonia (4 mL) was added to **20** (37 mg, 0.084 mmol). The mixture was stirred for 7 d at room temperature, and then for 4 d at 30 °C. The solvent was removed in vacuo, and the residue was purified by GPC (chloroform) to give **4a** (18 mg, 51%) as a colorless solid. ^1^H NMR (400 MHz, CDCl_3_) δ 5.36 (brs, 1H), 5.22 (brs, 1H), 2.32-2.25 (m, 1H), 2.15-2.07 (m, 1H), 1.95 (d, *J* = 11.4 Hz, 1H), 1.87-1.73 (m, 4H), 1.59-0.91 (m, 25H), 1.22 (s, 6H), 0.93 (d, *J* = 6.4 Hz, 3H), 0.91 (s, 3H), 0.64 (s, 3H); ^13^C NMR (150 MHz, CDCl_3_) δ 175.82, 71.65, 56.50, 55.93, 51.22, 43.54, 42.72, 40.46, 40.17, 37.42, 35.89, 35.80, 35.45, 34.97, 34.67, 32.75, 31.59, 30.03, 29.96, 29.76, 28.25, 27.39, 26.45, 24.17, 23.96, 20.78, 18.33, 12.02; HRMS calcd for C_28_H_49_NaO_2_ (M+Na)^+^ 454.3656, found 454.3649.

Compound **4b** was synthesized similarly. **4b**: ^1^H NMR (400 MHz, CDCl_3_) δ 5.38 (brs, 1H), 2.80 (d, *J* = 5.0 Hz, 3H), 2.23-2.19 (m, 1H), 2.08-2.02 (m, 1H), 1.94 (d, *J* = 11.4 Hz, 1H), 1.85-1.73 (m, 4H), 1.49-0.90 (m, 25H), 1.22 (s, 6H), 0.91 (d, *J* = 6.4 Hz, 3H), 0.91 (s, 3H), 0.63 (s, 3H); ^13^C NMR (150 MHz, CDCl_3_) δ 174.19, 71.66, 56.51, 55.97, 51.23, 43.55, 42.70, 40.47, 40.17, 37.43, 35.89, 35.80, 35.52, 34.97, 34.68, 33.55, 31.82, 30.03, 29.96, 29.77, 28.24, 27.40, 26.45, 26.30, 24.17, 23.97, 20.78, 18.35, 12.02; HRMS calcd for C_29_H_51_NaO_2_ (M + Na)^+^ 468.3796, found 468.3805.

**Synthesis of compound 21**: *O*-Benzylhydroxylamine hydrochloride (24 mg, 0.15 mmol), *N*,*N*-diisopropylethylamine (20 mg, 0.16 mmol) and 1-hydroxybenzotriazole (17 mg, 0.13 mmol) were successively added to a solution of **3** (49 mg, 0.11 mmol) in dry dichloromethane (10 mL) at room temperature. After 10 min, *N*,*N’*-dicyclohexylcarbodiimide (29 mg, 0.14 mmol) was added to it. The mixture was stirred for 23 h at room temperature, and then filtered. The filtrate was washed with 5% hydrochloric acid, and brine, dried over sodium sulfate, filtered, and concentrated. The residue was purified by silica gel column chromatography (ethyl acetate/*n*-hexane = 1:2) to give **21** (68 mg, quant.) as a colorless solid. ^1^H NMR (600 MHz, CDCl_3_) δ 7.85 (br, 1 H), 7.41-7.39 (m, 5 H), 4.92 (br, 2 H), 1.94-0.92 (m, 31 H), 1.22 (s, 6 H), 0.91 (s, 3 H), 0.88 (d, *J* = 4.8 Hz, 3 H), 0.62 (s, 3 H); ^13^C NMR (150 MHz, CDCl_3_) δ 171.44, 137.83, 129.24, 128.59, 128.18, 78.03, 71.66, 56.48, 55.88, 51.21, 43.53, 42.69, 40.45, 40.15, 37.41, 35.88, 35.78, 35.41, 34.95, 34.66, 31.43, 30.02, 29.93, 29.75, 29.66, 28.19, 27.38, 26.43, 24.15, 23.95, 20.76, 18.26, 12.02; HRMS calcd for C_35_H_56_NO_3_ (M + H)^+^ 538.4255, found 538.4241.

**Synthesis of compound 5b**: Compound **3** (20 mg, 0.047 mmol) in DMF (0.5 mL) was added to a solution of triethylamine (12 mg, 0.12 mmol) in DMF (0.5 mL). The mixture was stirred at 0 °C for 10 min, and then ethyl chloroformate (6 mg, 0.055 mmol) in DMF (0.5 mL) was added to it. The resulting mixture was stirred at 0 °C for 45 min, and a mixture of methoxyamine hydrochloride (4 mg, 0.052 mmol) and triethylamine (12 mg, 0.12 mmol) in DMF (1.0 mL) was added to it. Stirring was continued at room temperature for 4 h, then the solvent was removed in vacuo, and the residue was extracted with ethyl acetate. The organic layer was washed with brine, dried over sodium sulfate, filtered, and concentrated. The residue was purified by GPC (chloroform) to give **5b** (16 mg, 73%) as a colorless solid. ^1^H NMR (400 MHz, CDCl_3_) δ 7.98 (brs, 1H), 3.76 (s, 3H), 2.17 (brs, 1H), 1.94 (d, *J* = 10.5, 1H), 1.87-1.73 (m, 4H), 1.59-0.91 (m, H), 1.22 (s, 6H), 0.92 (d, *J* = 8.2 Hz, 3H), 0.91 (s, 3H), 0.64 (s, 3H) ; ^13^C NMR (150MHz, CDCl_3_) δ 171.64, 71.73, 64.56, 56.56, 55.95, 51.28, 43.60, 42.78, 40.52, 40.22, 37.48, 35.95, 35.85, 35.51, 35.03, 34.73, 31.45, 30.25, 30.09, 30.01, 29.83, 28.28, 27.45, 26.50, 24.22, 24.02, 20.84, 18.38, 12.08; HRMS calcd for C_29_H_51_NaO_3_ (M + Na)^+^ 484.3761, found 484.3762.

**Synthesis of compound 5a**: Palladium hydroxide (13 mg) was added to a solution of **21** (68 mg, 0.11 mmol) in dry methanol (15 mL). The mixture was stirred for 24 h at room temperature under a hydrogen atmosphere, then filtered, and the filtrate was concentrated. The residue was purified by silica gel column chromatography (ethyl acetate/*n*-hexane = 2:1, ethyl acetate, then, ethyl acetate/methanol = 20:1) to give **5a** (31 mg, 55%) as a colorless solid. ^1^H NMR (600 MHz, CDCl_3_) δ 5.40 (br, 1 H), 5.33 (br, 1 H), 2.28 (ddd, *J* = 15.6, 10.8, 4.2 Hz 1 H), 2.11 (ddd, *J* = 16.8, 10.8, 6.0 Hz 1 H), 1.95-1.92 (m, 1 H), 1.85-0.94 (m, 28 H), 1.22 (s, 6 H), 0.92 (d, *J* = 6.6 Hz, 3 H), 0.90 (s, 3 H), 0.63 (s, 3 H); ^13^C NMR (150 MHz, CDCl_3_) δ 176.04, 71.69, 56.52, 55.94, 51.23, 43.56, 42.73, 40.47, 40.18, 37.44, 35.90, 35.81, 35.46, 34.98, 34.69, 32.77, 31.61, 30.04, 29.97, 29.78, 28.27, 27.41, 26.46, 24.18, 23.98, 20.80, 18.35, 12.04; Anal. calcd for C_28_H_49_NO_3_: C, 75.12; H, 11.03; N, 3.15, found: C, 76.35; H, 11.04; N, 3.20.

**Synthesis of compound 6**: 4-Dimethylaminopyridine (23 mg, 0.19 mmol), cyanamide (19 mg, 0.45 mmol), 1-(3-dimethylaminopropyl)-3-ethylcarbodiimide hydrochloride (42 mg, 0.22 mmol) and *N*,*N*-diisopropylethylamine (39 mg, 0.30 mmol) were successively added to a solution of **3** (59 mg, 0.14 mmol) in dry dichloromethane (5 mL). The mixture was stirred for 18 h at room temperature under an argon atmosphere, then diluted with dichloromethane, washed with 2 M hydrochloric acid and brine, dried over sodium sulfate, filtered, and concentrated. The residue was purified by silica gel flash column chromatography (chloroform/methanol = 10:1) to give **6** (5 mg, 89%) as a colorless oil. ^1^H NMR (600 MHz, pyridine-*d_5_*) δ 2.65 (br, 1 H), 2.52 (br, 1 H), 2.01-1.95 (m, 1 H), 1.86-1.72 (m, 6 H), 1.65-0.85 (m, 22 H), 1.44 (s, 6 H), 0.92 (s, 3 H), 0.88 (d, *J* = 6.0 Hz, 3 H), 0.56 (s, 3 H); ^13^C NMR (150 MHz, CDCl_3_) δ 174.15, 108.26, 71.57, 56.39, 55.76, 51.03, 43.46, 42.64, 40.36, 40.05, 37.34, 35.82, 35.70, 35.21, 34.85, 34.57, 32.21, 30.49, 29.68, 29.62, 29.51, 28.08, 27.30, 26.35, 24.07, 23.85, 20.68, 18.09, 11.90; HRMS calcd for C_29_H_47_N_2_O_2_ (M-H)^-^ 455.3643, found 455.3627.

**Synthesis of compound 22**: L-Glycine methyl ester hydrochloride (8 mg, 0.06 mmol) and *N*-methylmorpholine (13 mg, 0.12 mmol) were added to a solution of **3** (20 mg, 0.047 mmol) in dry dichloromethane (8 mL). 1-(3-Dimethylaminopropyl)-3-ethylcarbodiimide hydrochloride (12 mg, 0.06 mmol) was added to the mixture under an argon atmosphere. The resulting mixture was stirred for 24 h at room temperature, then diluted with dichloromethane, washed with 2 M hydrochloric acid, and brine, dried over sodium sulfate, filtered, and concentrated. The residue was purified by silica gel column chromatography (dichloromethane/methanol = 19:1) to give **22a** (19 mg, 83%) as a colorless solid. ^1^H NMR (600 MHz, CDCl_3_) δ 5.91 (br, 1 H), 4.05 (d, *J* = 5.4 Hz, 2 H), 3.77 (s, 3 H), 2.30 (ddd, *J* = 15.6, 10.8, 5.4 Hz 1 H), 2.13 (ddd, *J* = 15.0, 10.2, 6.0 Hz 1 H), 1.96-1.94 (m, 1 H), 1.87-1.73 (m, 5 H), 1.56-0.96 (m, 23 H), 1.22 (s, 6 H), 0.92 (d, *J* = 6.6 Hz, 3 H), 0.91 (s, 3 H), 0.63 (s, 3 H); ^13^C NMR (150 MHz, CDCl_3_) δ 173.58, 170.52, 71.55, 56.40, 55.85, 52.19, 51.13, 43.45, 42.61, 41.07, 40.36, 40.07, 37.33, 35.79, 35.70, 35.36, 34.87, 34.58, 33.14, 31.46, 29.93, 29.86, 29.67, 28.13, 27.30, 26.35, 24.07, 23.87, 20.68, 18.23, 11.92; HRMS calcd for C_31_H_53_NNaO_4_ (M + Na)^+^ 526.3867, found 526.3860.

Compounds **22b** and **22c** were synthesized similarly. **22b**: ^1^H NMR (600 MHz, CDCl_3_) δ 6.01 (t, *J* = 6.6 Hz, 1 H), 3.70 (s, 3 H), 3.51 (q, *J* = 6.0 Hz, 2 H), 2.54 (t, *J* = 6.0 Hz, 2 H), 2.24-2.18 (m, 1 H), 2.07-2.01 (m, 1 H), 1.95-1.92 (m, 1 H), 1.87-1.73 (m, 4 H), 1.56-0.96 (m, 24 H), 1.22 (s, 6 H), 0.91 (s, 3 H), 0.90 (d, *J* = 6.0 Hz, 3 H), 0.63 (s, 3 H); ^13^C NMR (150 MHz, CDCl_3_) δ 173.64, 173.32, 71.67, 56.50, 55.96, 51.83, 51.22, 43.55, 42.70, 40.46, 40.17, 37.43, 35.89, 35.80, 35.47, 34.97, 34.67, 33.81, 33.62, 31.70, 30.02, 29.95, 29.77, 28.24, 27.40, 26.45, 24.17, 23.96, 20.78, 18.33, 12.01; HRMS calcd for C_32_H_55_NNaO_4_ (M + Na)^+^ 540.4023, found 540.4033. **22c**: ^1^H NMR (600 MHz, CDCl_3_) δ 5.63 (br, 1 H), 3.68 (s, 3 H), 3.29 (q, *J* = 7.2 Hz, 2 H), 2.37 (t, *J* = 7.2 Hz, 2 H), 2.22 (ddd, *J* = 15.6, 10.2, 4.2 Hz, 1 H), 2.04 (ddd, *J* = 15.6, 10.2, 6.0 Hz, 1 H), 1.94-1.02 (m, 1 H), 1.84 (q, *J* = 7.2 Hz, 2 H), 1.77-1.73 (m, 3 H), 1.57-0.93 (m, 25 H), 1.22 (s, 6 H), 0.91 (d, *J* = 6.6 Hz, 3 H), 0.91 (s, 3 H), 0.63 (s, 3 H); ^13^C NMR (150 MHz, CDCl_3_) δ 173.98, 173.80, 71.66, 56.49, 55.95, 51.75, 51.21, 43.54, 42.69, 40.45, 40.16, 38.92, 37.42, 35.88, 35.79, 35.48, 34.96, 34.66, 33.61, 31.77, 31.46, 30.01, 29.94, 29.76, 28.24, 27.39, 26.44, 24.54, 24.16, 23.95, 20.77, 18.34, 12.00; HRMS calcd for C_33_H_57_NNaO_4_ (M + Na)^+^ 554. 4180, found 554.4179.

**Synthesis of compound 7**: 15% *w/v* aqueous sodium hydroxide (1 mL) was added to a solution of **22a** (8 mg, 0.035 mmol) in ethanol (5 mL), and the mixture was stirred for 3 h at room temperature. Ethanol was removed in vacuo, and the solution was acidified with conc. hydrochloric acid until a precipitate was formed. This was collected and washed with water to give **7a** (13 mg, 77%) as a colorless solid. ^1^H NMR (600 MHz, CD_3_OD) δ 3.86 (s, 2 H), 2.30 (ddd, *J* = 13.8, 10.2, 5.4 Hz, 1 H), 2.15 (ddd, *J* = 13.8, 9.6, 6.6 Hz, 1 H), 2.00-1.98 (m, 1 H), 1.91-1.85 (m, 2 H), 1.82-1.76 (m, 2 H), 1.60-0.98 (m, 24 H), 1.17 (s, 6 H), 0.96 (d, *J* = 6.0 Hz, 3 H), 0.93 (s, 3 H), 0.68 (s, 3 H); ^13^C NMR (150 MHz, CD_3_OD) δ 177.10, 173.50, 72.01, 57.95, 57.46, 52.08, 49.56, 45.15, 43.91, 42.03, 41.90, 41.57, 38.69, 37.27, 36.83, 36.22, 35.82, 33.81, 33.08, 31.00, 29.93, 29.91, 29.25, 28.63, 27.75, 25.30, 24.54, 21.94, 18.83, 12.47; HRMS calcd for C_30_H_51_NNaO_4_ (M + Na)^+^ 512.3710, found 512.3707.

Compounds **7b** and **7c** were synthesized similarly. **7b**: ^1^H NMR (600 MHz, CD_3_OD) δ 3.39 (t, *J* = 6.6 Hz, 2 H), 2.48 (t, *J* = 6.6 Hz, 2 H), 2.21-2.19 (m, 1 H), 2.10-2.05 (m, 1 H), 2.00-1.98 (m, 1 H), 1.91-1.87 (m, 2 H), 1.78-1.73 (m, 2 H), 1.59-1.49 (m, 4 H), 1.40-0.96 (m, 26 H), 0.94 (d, *J* = 7.2 Hz, 3 H), 0.93 (s, 3 H), 0.67 (s, 3 H); ^13^C NMR (150 MHz, CD_3_OD) δ 176.87, 175.48, 72.00, 57.94, 57.43, 52.08, 45.14, 43.90, 41.89, 41.55, 38.68, 37.26, 36.85, 36.42, 36.21, 35.81, 34.77, 34.00, 33.28, 30.99, 29.92, 29.90, 29.28, 28.62, 27.74, 25.28, 24.53, 21.93, 18.81, 12.45; HRMS calcd for C_31_H_53_NNaO_4_ (M + Na)^+^ 526.3867, found 526.3862. **7c**: ^1^H NMR (600 MHz, CDCl_3_) δ 5.63 (br, 1 H), 3.68 (s, 3 H), 3.29 (q, *J* = 7.2 Hz, 2 H), 2.37 (t, *J* = 7.2 Hz, 2 H), 2.22 (ddd, *J* = 15.6, 10.2, 4.2 Hz, 1 H), 2.04 (ddd, *J* = 15.6, 10.2, 6.0 Hz, 1 H), 1.94-1.02 (m, 1 H), 1.84 (q, *J* = 7.2 Hz, 2 H), 1.77-1.73 (m, 3 H), 1.57-0.93 (m, 25 H), 1.22 (s, 6 H), 0.91 (d, *J* = 6.6 Hz, 3 H), 0.91 (s, 3 H), 0.63 (s, 3 H); ^13^C NMR (150 MHz, CDCl_3_) δ 173.98, 173.80, 71.66, 56.49, 55.95, 51.75, 51.21, 43.54, 42.69, 40.45, 40.16, 38.92, 37.42, 35.88, 35.79, 35.48, 34.96, 34.66, 33.61, 31.77, 31.46, 30.01, 29.94, 29.76, 28.24, 27.39, 26.44, 24.54, 24.16, 23.95, 20.77, 18.34, 12.00; HRMS calcd for C_33_H_57_NNaO_4_ (M + Na)^+^ 554. 4180, found 554.4179.

**Synthesis of compound 8**: 4-(4,6-Dimethoxy-1,3,5-triazin-2-yl)-4-methylmorpholinium chloride (123 mg, 0.44 mmol) and triethylamine (437 mg, 4.32 mmol) were added to a solution of **3** (75 mg, 0.17 mmol) in dry *N*,*N*-dimethylformamide (8 mL). The mixture was stirred for 10 min at room temperature under an argon atmosphere, and aminomethanesulfuric acid (132.2 mg, 1.190 mmol) was added to it. The mixture was stirred for 20 h at room temperature, then filtered, and the solvent was removed in vacuo. The residue was extracted with ethyl acetate and water. The water layer was cooled to 0 °C, and conc. hydrochloric acid (5.0 mL) was added. The resulting precipitate was collected, and washed with water to give **8a** (67 mg, 74%) as a pale yellow solid. ^1^H NMR (600 MHz, CD_3_OD) δ 4.16 (q, *J* = 13.2 Hz, 2 H), 2.18 (br, 1 H), 2.02 (br, 1 H), 1.85 (d, *J* = 11.4 Hz, 1 H), 1.75 (m, 2 H), 1.63 (d, *J* = 13.8 Hz, 2 H), 1.44-0.84 (m, 24 H), 1.03 (s, 6 H), 0.81 (d, *J* = 6.0 Hz, 3 H), 0.79 (s, 3 H), 0.53 (s, 3 H); HRMS calcd for C_29_H_50_NO_5_S (M-H)^-^ 524.3415, found 524.3404.

Compound **8b** was synthesized similarly. **8b**: ^1^H NMR (600 MHz, Pyrdine-*d_5_*) δ 4.23 (t, *J* = 5.4 Hz, 2 H), 3.48 (t, *J* = 5.4 Hz, 2 H), 2.40 (ddd, *J* = 14.4, 10.2, 4.2 Hz, 1 H), 2.25 (ddd, *J* = 16.2, 10.2, 6.0 Hz, 1 H), 2.00-1.98 (m, 1 H), 1.82-0.96 (m, 28 H), 1.42 (s, 6 H), 0.90 (s, 3 H), 0.85 (d, *J* = 6.0 Hz, 3 H), 0.53 (s, 3 H); ^13^C NMR (150 MHz, Pyrdine-*d_5_*) δ 174.07, 70.15, 56.46, 56.19, 52.01, 51.70, 44.00, 42.79, 40.71, 40.33, 37.92, 36.52, 36.41, 35.96, 35.73, 35.45, 34.90, 33.78, 32.26, 30.78, 30.74, 30.33, 28.35, 27.80, 26.75, 24.36, 24.24, 21.08, 18.55, 12.19; HRMS calcd for C_30_H_52_NO_5_S (M-H)^-^ 538.3572, found 528.3576.

### 4.3. HL-60 Cell Differentiation Assay 

HL-60 cells were cultured in RPMI-1640 medium supplemented with 5% FBS and penicillin G and streptomycin at 37 °C under 5% CO_2_ in air [14]. The cells were diluted to 8.0 × 10^4^ cells/mL with RPMI-1640 (5% FBS), and an ethanol solution of a test compound was added to give 10^−9^ to 10^−6^ M final concentration. Control cells were treated with the same volume of ethanol alone. 1α,25(OH)_2_D_3_ was always assayed at the same time as a positive control. The cells were incubated at 37 °C under 5% CO_2_ in air for 4 days. The percentage of differentiated cells was determined by nitro-blue tetrazolium (NBT) reduction assay. Cells were incubated at 37 °C for 20 min in RPMI-1640 (5% FBS) and an equal volume of phosphate-buffered saline (PBS) containing NBT (0.2%) and 12-*O*-tetradecanoylphorbol 13-acetate (TPA; 200 ng/mL). The percentage of cells containing blue-black formazan was determined in a minimum of 200 cells. All experiments were done in triplicate.

### 4.4. Transactivation Assay

Human embryonic kidney HEK293 cells (RIKEN Cell Bank, Tsukuba, Japan) were cultured in Dulbecco’s modified Eagle’s medium containing 5% FBS, 100 U/mL penicillin, and 0.1 mg/mL streptomycin (Nacalai Tesque, Kyoto, Japan). Transfections used 15 ng of pCMX-hVDR, 50 ng of TK-Spp × 3-LUC reporter plasmid, and 10 ng of pCMX-β-galactosidase for each well of a 96-well plate, and were performed by the calcium phosphate coprecipitation assay as described previously [13]. Eight hours after transfection, test compounds were added. Cells were harvested after 16–24 h and were assayed for luciferase and β-galactosidase activity using a luminometer and a microplate reader (Molecular Devices, Sunnyvale, CA, USA). Luciferase data were normalized to the internal β-galactosidase control. All experiments were done in triplicate.

### 4.5. X-ray Crystallographic Analysis

Crystals of VDR complexes were prepared according to the method of Vanhooke et al. [16] with some modifications. The rat VDR LBD (residues 116–423, Δ165–211) was cloned as an N-terminal His6-tagged fusion protein into the pET14b expression vector and overproduced in *Escherichia coli* C41. The cells were grown at 37 °C in LB medium (including ampicillin 100 mg/L) and subsequently induced for 6 h with 15 µM iso-propyl-β-d-thiogalactopyranoside (IPTG) at 23 °C. The purification procedure included affinity chromatography on a Ni-NTA column, followed by dialysis and cation-exchange chromatography (SP-Sepharose). After tag removal by thrombin digestion, the protease was removed by filtration through a HiTrap benzamidine column and the protein was further purified by gel filtration on a Super-dex200 column. The purity and homogeneity of the rVDR LBD were assessed by SDS-PAGE.

Purified rVDR LBD solution was concentrated to about 0.75 mg/mL by ultrafiltration. To an aliquot (800 µL) of the protein solution a ligand was added (approx. 10 equiv). Then the solution was further concentrated to about 1/8, and a solution (25 mM Tris-HCl, pH 8.0; 50 mM NaCl; 10 mM DTT; 0.02% NaN_3_) of coactivator peptide (H_2_N-KNHPMLMNLLKDN-CONH_2_) derived from DRIP205 was added. This solution of VDR/ligand/peptide was allowed to crystallize by the vapor diffusion method using a series of precipitant solutions containing 0.2 M potassium citrate tribasic monohydrate, 20% (*w/v*) PEG3350. Droplets for crystallization were prepared by mixing 2 μL of complex solution and 1 μL precipitant solution, and equilibrated against 500 μL of precipitant solution at 20 °C.

Prior to diffraction data collection, crystals were soaked in a cryoprotectant solution containing 0.2 M potassium citrate tribasic monohydrate, 20% (*w/v*) PEG3350, and 17–20% ethylene glycol. Diffraction data sets were collected at 100 K in a stream of nitrogen gas at beamline BL-17A of KEK-PF (Tsukuba, Japan). Reflections were recorded with an oscillation range per image of 1.0°. Diffraction data were indexed, integrated, and scaled using the program HKL2000 (HKL Research Inc., Charlottesville, VA, USA). The structures were solved by molecular replacement with the program Phaser in Phenix [17], using the rat VDR LBD coordinates (PDB code: 2ZLC), and finalized sets of atomic coordinates were obtained after iterative rounds of model modification with the program COOT [18] and refinement with REFMAC [19]. The coordinates and structure factors have been deposited in the Protein Data Bank (Entry ID: 7VQP).

## Data Availability

Not applicable.

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
