# Peer review of "Lithocholic Acid Amides as Potent Vitamin D Receptor Agonists"

_biomolecules, 2022, doi:10.3390/biom12010130_

Round 1
Reviewer 1 Report
Major Comments:
1. An author may provide the PEC50 value in the manuscript.
2. The author may need to write more about the results and conclusion part
Why author didn’t take the vitamin D compound for cell line study.
3. The author may clarify how the synthesized compound would be better than the known active Vitamin D compound.
4.The X-ray structure part might be confused so the author can write precisely.
Author Response
On the basis of the reviewers’ comments, we revised our manuscript. The revised part was colored in red in the revised manuscript. Our responses toward the comments given by the reviewer #1 are listed below. The manuscript was checked by a native English speaker.
Comment #1: An author may provide the PEC50 value in the manuscript.
Our response: Thank you for your comment. We added the EC50 values of the assay in Table 1 (new). As for HL-60 assay, we reexamined the experiments with n = 3. The data for compound 4b are different from those in the previous version. We confirmed that the new data are correct.
Comment #2: The author may need to write more about the results and conclusion part. Why author didn’t take the vitamin D compound for cell line study.
Our response: Thank you for your comment. We added the transactivation assay of selected compounds in Figure 4 (new) and Table 1 (new), and added the results, discussion and experimental section about the transactivation activity. We added two authors who employed the additional experiments.
Comment #3: The author may clarify how the synthesized compound would be better than the known active Vitamin D compound.
Our response: Thank you for your comment. Compound 3 is potent vitamin D derivatives with non-secosteroidal structure, that has advantage in the chemical properties, compared to the secosteroid-type vitamin D derivatives, and was reported to have lower calcemic activity than 1α,25(OH)2D3 (ref 9). These properties of compound 3 would be favorable for clinical application, while compound 3 was eliminated very quickly from the serum in mice, and the pharmacophore properties should be improved. The results in this study, the amide derivatives were indicated to be candidate compounds with potent activity and better pharmacokinetic properties. Detailed pharmacokinetic properties are now on going. These discussion is described in the conclusion section.
Comment #4: The X-ray structure part might be confused so the author can write precisely.
Our response: Thank you for your comment. We modified the part of results of X-ray crystallographic analysis. We added the distance of hydrogen bondings.
(The revised sentences, the revised points are shown by underline.)
Figure 5a shows the overall structure of the VDR LBD complex with 7b; it is similar to those previously reported for VDR LBD complexes with other lithocholic acid derivatives [8,10]. The interactions of compound 7b with amino acid residues of the VDR LBD (Figure 5b) are compared with those of 3 (Dcha-20) in Figure 5c. The hydroxyl group in the 3-substituent of 7b forms direct hydrogen bonds with two histidine residues, His301 (O···N distance: 2.79 Å) and His393 (O···N distance: 2.68 Å). This is the same as in the case of 3 (Dcha-20), in which the O···N distances were 2.80 Å for His301 and 2.66 Å for His393, whereas LCA (2) forms indirect hydrogen bonds with these amino acid residues via a water molecule. The direct interactions of the hydroxy group in the 3-substituent with two histidines may contribute to the potent activity of 7b and 3. The carboxyl group of compound 3 (Dcha-20) formed hydrogen bonds with the phenolic hydroxyl group of Tyr143 (O··O distance: 2.81 Å) in helix 1 and the hydroxymethyl group of Ser274 (O··O distance: 3.13 Å) in helices 4/5, whereas the amide group of 7b did not form a hydrogen bond with any amino acid residue. Instead, the terminal carboxyl group of 7b formed hydrogen bonds with Arg270 (O···N distance: 2.82 Å) and the backbone amide bond of Tyr143 (O···N distance: 2.91 Å). Similar hydrogen bond formation with these amino acid residues of the VDR LBD was observed in secosteroid derivatives bearing a hydroxylated substituent at the 2-position of the cyclohexane ring.
Reviewer 2 Report
The manuscript is well written and open new avenues of LCA derivatives as VDR agonists. Minor points need to be addressed to support their findings.
The distances in Å of the hydrogen bonds in Figure 4 should be included.
Reference of a publication describing hydroxylated LCA forming similar hydrogen bonds than 1,25D3 should be included (Gonzalez et al., 2021).
The authors should discuss the relative low number of water molecules (49).
Transactivation assay of VDR overexpressing cells should be added to the characterization of the various compounds.
Author Response
On the basis of the reviewers’ comments, we revised our manuscript. The revised part was colored in red in the revised manuscript. Our responses toward the comments given by the reviewer #2 are listed below. The manuscript was checked by a native English speaker.
Comment #1: The distances in AÌŠ of the hydrogen bonds in Figure 4 should be included.
Our response: Thank you for your comment. Addition of distances of hydrogen bondings made the figures unclear. Therefore, we added them in the text.
(The revised sentences, the revised points are shown by underline.)
Figure 5a shows the overall structure of the VDR LBD complex with 7b; it is similar to those previously reported for VDR LBD complexes with other lithocholic acid derivatives [8,10]. The interactions of compound 7b with amino acid residues of the VDR LBD (Figure 5b) are compared with those of 3 (Dcha-20) in Figure 5c. The hydroxyl group in the 3-substituent of 7b forms direct hydrogen bonds with two histidine residues, His301 (O···N distance: 2.79 Å) and His393 (O···N distance: 2.68 Å). This is the same as in the case of 3 (Dcha-20), in which the O···N distances were 2.80 Å for His301 and 2.66 Å for His393, whereas LCA (2) forms indirect hydrogen bonds with these amino acid residues via a water molecule. The direct interactions of the hydroxy group in the 3-substituent with two histidines may contribute to the potent activity of 7b and 3. The carboxyl group of compound 3 (Dcha-20) formed hydrogen bonds with the phenolic hydroxyl group of Tyr143 (O··O distance: 2.81 Å) in helix 1 and the hydroxymethyl group of Ser274 (O··O distance: 3.13 Å) in helices 4/5, whereas the amide group of 7b did not form a hydrogen bond with any amino acid residue. Instead, the terminal carboxyl group of 7b formed hydrogen bonds with Arg270 (O···N distance: 2.82 Å) and the backbone amide bond of Tyr143 (O···N distance: 2.91 Å). Similar hydrogen bond formation with these amino acid residues of the VDR LBD was observed in secosteroid derivatives bearing a hydroxylated substituent at the 2-position of the cyclohexane ring.
Comment #2: Reference of a publication describing hydroxylated LCA forming similar hydrogen bonds than 1,25D3 should be included (Gonzalez et al., 2021).
Our response: Thank you for your comment. We added the above reference as ref. 13.
Comment #3: The authors should discuss the relative low number of water molecules (49).
Our response: Thank you for your comment. We added water molecules in a conservative way, where only spherical densities with a reasonable distance from H-bond partner were assigned as water molecule. In fact, adding more did not improve the Free R value. So we think our current model is reasonable at least in terms of refinement. Even so, as you say, the number is relatively low. We have checked our diffraction images and analyses intensities with Xtriage of Phenix but did not find anything particularly unusual. Thus, we simply added the following sentence to acknowledge
this fact.
(Added sentence) The electron density map clearly shows the VDR LBD, the coactivator peptide, the ligand and a relatively low number of water molecules.
Comment #4: Transactivation assay of VDR overexpressing cells should be added to the characterization of the various compounds.
Our response: Thank you for your comment. We added the transactivation assay of selected compounds in Figure 4 (new) and Table 1 (new), and added the results, discussion and experimental section about the transactivation activity. We added two authors who employed the additional experiments.
Reviewer 3 Report
The paper by Ayana Yoshihara et al. describes the methods of synthesis of a series of amide derivatives of litocholic acid (LCA), and some evaluation of their biological activity. LCA is a metabolite of bile acids, and it is a week agonist of the vitamin D receptor (VDR). Some chemical modifications introduced into the structure of LCA significantly enhance binding of the compound to VDR and transcriptional activation of this receptor. The group of authors of this paper has published recently another paper in which they described a LCA derivative, which is a very potent VDR agonist. Now they further modified the former derivative, and estimated how these new modifications influence biological activity.
In contrast to the previous paper, this time only one biological assay has been used. This was so called "NBT reduction assay" in which the authors tested whether their new compounds are able to activate monocyte-specific enzymes in HL60 cells or not.
The authors also tried to crystalize their new compounds with the VDR ligand binding domain and with a coacticator peptide, but they were succesful only in case of one compound. Therefore, in my opinion, the paper does not indicate that the NBT-reducing activity in HL60 cells is indeed mediated by VDR.
I would suggest to include VDR competitive binding assay, at least for the most active compounds, in order to show that indeed they are able to bind VDR.
Moreover, the presentation of data in Figure 3 is confusing. The same symbols are used for various compounds, what hinders the reading. Since almost all curves presented in the graphs have sigmoidal shapes, I would suggest adding a table with EC50 values of each compound.
Author Response
On the basis of the reviewers’ comments, we revised our manuscript. The revised part was colored in red in the revised manuscript. Our responses toward the comments given by the reviewer #3 are listed below. The manuscript was checked by a native English speaker.
Comment #1: In contrast to the previous paper, this time only one biological assay has been used. This was so called "NBT reduction assay" in which the authors tested whether their new compounds are able to activate monocyte-specific enzymes in HL60 cells or not.
The authors also tried to crystalize their new compounds with the VDR ligand binding domain and with a coactivator peptide, but they were successful only in case of one compound. Therefore, in my opinion, the paper does not indicate that the NBT-reducing activity in HL60 cells is indeed mediated by VDR.
I would suggest to include VDR competitive binding assay, at least for the most active compounds, in order to show that indeed they are able to bind VDR.
Our response: Thank you for your comment. In our laboratory, it is difficult to examined the VDR binding assay, Instead, we examined the transactivation assay, and added the data in Figure 4 (new) and Table 1 (new). We added two authors who employed the additional experiments.
Comment #2: Moreover, the presentation of data in Figure 3 is confusing. The same symbols are used for various compounds, what hinders the reading. Since almost all curves presented in the graphs have sigmoidal shapes, I would suggest adding a table with EC50 values of each compound.
Our response: Thank you for your comment. The symbol/color in Figure 3 was revised. We added the EC50 values of the assay in Table 1 (new).
Round 2
Reviewer 1 Report
The Authors have revised and Updated the manuscript as per my suggestion. So, I kindly recommend this article for publication.
Author response:
Thanks for your review.
Reviewer 3 Report
Thank you for correcting your paper.
Author response:
Thanks for your review.